# Roles of OmpA in Type III Secretion System-Mediated Virulence of Enterohemorrhagic *Escherichia coli*

**DOI:** 10.3390/pathogens10111496

**Published:** 2021-11-17

**Authors:** Hidetada Hirakawa, Kazutomo Suzue, Ayako Takita, Haruyoshi Tomita

**Affiliations:** 1Department of Bacteriology, Graduate School of Medicine, Gunma University, Maebashi 371-8511, Gunma, Japan; takita626@gunma-u.ac.jp (A.T.); tomitaha@gunma-u.ac.jp (H.T.); 2Department of Infectious Diseases and Host Defense, Graduate School of Medicine, Gunma University, Maebashi 371-8511, Gunma, Japan; 3Laboratory of Bacterial Drug Resistance, Graduate School of Medicine, Gunma University, Maebashi 371-8511, Gunma, Japan

**Keywords:** virulence, pathogenicity, type III secretion system, gastrointestinal infection, outer membrane protein, enteric pathogen

## Abstract

Outer membrane proteins are commonly produced by gram-negative bacteria, and they have diverse functions. A subgroup of proteins, which includes OmpA, OmpW and OmpX, is often involved in bacterial pathogenesis. Here we show that OmpA, rather than OmpW or OmpX, contributes to the virulence of enterohemorrhagic *Esch**erichia coli* (EHEC) through its type III secretion system (T3SS). Deletion of *ompA* decreased secretion of the T3SS proteins EspA and EspB; however, the expression level of the LEE genes that encode a set of T3SS proteins did not decrease. The *ompA* mutant had less abilities to form A/E lesions in host epithelial cells and lyse human red blood cells than the parent strain. Moreover, the virulence of an *ompA* mutant of *Citrobacter rodentium* (traditionally used to estimate T3SS-associated virulence in mice) was attenuated. Mice infected with the *ompA* mutant survived longer than those infected with the parent strain. Furthermore, mice infected with *ompA* developed symptoms of diarrhea more slowly than mice infected with the parent strain. Altogether, these results suggest that OmpA sustains the activity of the T3SS and is required for optimal virulence in EHEC. This work expands the roles of outer membrane proteins in bacterial pathogenesis.

## 1. Introduction

Enterohemorrhagic *Escherichia coli* (EHEC) is one of the most common foodborne pathogens. It can cause diarrhea and colitis. Moreover, infections by EHEC can also lead to severe complications, such as hemolytic-uremic syndrome (HUS), leading to acute kidney failure and acute encephalopathy [1]. The two major types of proteins responsible for the pathogenicity of EHEC are Shiga toxins and effector proteins. Shiga toxins inhibit protein synthesis by impairing the ribosomal activity of eukaryotic host cells, resulting in cell death [2]. Effector proteins bind to certain host proteins in intestinal epithelial cells, which then induce attaching and effacing (A/E) lesions [3,4]. A/E lesions are characterized by bacterial attachment to the host plasma membrane, the destruction of gut epithelial microvilli, and accumulation of actin in the host cells [5]. The effector proteins are secreted via a transport complex with a needle-like structure termed the type III secretion system (T3SS). They target host cells via translocator proteins, such as EspA, EspB and EspD [4]. EspA forms a polymeric filamentous structure that enables the translocation of effector proteins to the host cells [6,7]. EspB and EspD are delivered via EspA and then they form a pore structure complex on the host cell membrane [8]. Intimin receptor (Tir) is delivered into the host cells as the first effector protein, and re-translocated onto the host plasma membrane, which enables bacteria to attach to the host cells [5]. EspB also acts as an effector by binding to several host proteins (such as α-catenin and myosin) and affects their activities [9,10].

OmpA, OmpW, and OmpX are the major members of the family of outer membrane proteins composed of an eight-stranded β-barrel with membrane-spanning regions [11]. These proteins are highly conserved in many gram-negative bacteria and their roles in bacterial pathogenesis have been characterized in several pathogens including *E. coli*, *Salmonella enterica*, *Pseudomonas aeruginosa*, *Vibrio cholerae*, *Klebsiella pneumoniae* and *Yersinia pestis* [12,13,14,15,16,17,18,19,20,21,22]. OmpA and OmpX have been implicated in the pathogenesis of several strains of *E. coli*. Deletion of the *ompA* gene of meningitic strains results in decreases in bacterial invasion into brain microvascular cells and survival within macrophages [12,13]. The *ompA* mutants in EHEC and uropathogenic *E. coli* (UPEC) exhibit defective adhesion to colon epithelial cells and colonization of the mouse bladder, respectively [14,15]. Meanwhile, compared to the wild-type, the *ompX* mutants in a pig lung disease-related strain and UPEC exhibit lower mouse mortality, and less flagellar production and colonization within the mouse kidney, respectively [16,17].

In this study, we investigated the roles of OmpA, OmpW, and OmpX in *E. coli* pathogenesis, focusing on EHEC. We show that OmpA, but not OmpW and OmpX, contributes to the T3SS-associated pathogenicity of EHEC.

## 2. Results

### 2.1. Deletion of ompA, but Not ompW and ompX, Decreases Extracellular Levels of the Type III Secretory Proteins EspA and EspB

We constructed in-frame deletion mutants of the *ompA*, *ompW*, and *ompX* genes. The gene deletions were confirmed by PCR assays. The DNA fragments were amplified from the chromosomal DNA templates of the EHEC O157 parent strain and its mutants with the delta1 and delta4 primer pairs shown in the Materials and Methods section. When the *ompA*, *ompW*, and *ompX* genes are intact, the predicted sizes of the PCR products are approximately 1950, 1550, and 1430 base pairs, respectively. If these genes are deleted, those of the resulting fragments are shortened to approximately 900 base pairs. We observed the shortened PCR fragments corresponding to gene deletions from the chromosomal DNA templates of the *ompA*, *ompW*, and *ompX* mutants (Figure 1A–C).

To investigate the role of OmpA, OmpW, and OmpX in EHEC pathogenesis, we first determined the levels of the EspB protein secreted via the T3SS in the *ompA*, *ompW*, and *ompX* mutants, then compared them to that of the parent strain by western blotting using an EspB antiserum. Only the *ompA* mutant secreted EspB at a level lower than that of the parent strain (Figure 2A). In contrast, the intracellular levels of EspB in the parent strain and the *ompA* mutant did not differ (Figure 2B). We confirmed that the decreased level of EspB secretion by *ompA* deletion was restored when *ompA* was expressed heterologously by the introduction of the pTrc99KompA plasmid (Figure 2C). Similarly, the level of secretion of EspA, the other protein secreted via the T3SS, was also lower in the *ompA* mutant compared to that of the parent strain (Figure 3A), and the introduction of the pTrc99KompA plasmid in this mutant elevated EspA secretion to the parent level (Figure 3B). The genes encoding T3SS proteins are clustered in five operons termed “the LEE operon” [23]. The *espB* and *espA* genes are transcribed in the same operon. We measured the transcript levels of *ler*, *escJ*, *escV*, *espA*, and *tir* from each operon. Figure 4 shows that the transcript levels of these genes in the *ompA* mutant and the parent strain were similar (Figure 4). Overall, these combined results suggest that the *ompA* gene contributes to the secretion, but not the expression of EspB and EspA.

The OmpA protein interacts with TolB, which is part of the Tol/Pal protein complex [24]; thus, we expected deletion of *ompA* to affect the function of Tol/Pal. Our previous study showed that deletion of the *tol/pal* genes reduces EspB secretion [25]. Therefore, we tested whether the decreased secretion of EspB by the *ompA* deletion is associated with a perturbation in the Tol/Pal protein complex. Consistent with the result of our previous study, the *tolB* mutant secreted less EspB than the parent strain, and deleting *ompA* in the *tolB* mutant, lacking TolPal activity, resulted in a further reduction in EspB secretion (Figure 5). These results suggest that the decreased secretion of EspB by the *ompA* deletion is not associated with Tol/Pal activity.

### 2.2. Deletion of ompA Reduces Abilities of EHEC to Form A/E Lesions and to Lyse Red Blood Cells (RBCs) Associated with the T3SS Activity

The effector proteins induce the formation of A/E lesions in the host epithelial cells. Reduction in the EspA and EspB secretions through the deletion of *ompA* may decrease the formation of A/E lesions. To test this hypothesis, we compared the ability of the *ompA* mutant to form A/E lesions with that of the parent strain. HeLa cells were infected with the parent strain or the *ompA* mutant, then we investigated actin accumulation with bacterial cells. We observed some actin accumulation in the HeLa cells infected with the parent strain. On the other hand, the frequency of actin accumulation was relatively low in those infected with the *ompA* mutant (Figure 6A). We also examined hemolytic activities of the *ompA* mutant and the parent strain with human RBCs. The activity of the *ompA* mutant was five-fold lower than that of the parent strain (Figure 6B). These results indicate that the *ompA* gene contributes to A/E lesion formation in host epithelial cells and hemolytic activity associated with the T3SS activity.

### 2.3. Neither ompA, Nor ompW, Nor ompX Deletion Decreases the Level of Shiga Toxins

Shiga toxins comprise the other subset of proteins responsible for EHEC pathogenesis. We used latex agglutination assays to estimate the levels of these proteins in the parent and its *ompA*, *ompW*, and *ompX* mutant strains. The agglutination titers of both Stx1 and Stx2 did not differ among all the strains (Table 1). Moreover, the transcript levels of *stx1* and *stx2* in the *ompA* mutant did not differ from those in the parent strain (Figure 4). Therefore, the *ompA*, *ompW*, and *ompX* genes are unlikely to contribute to the production of Shiga toxins.

### 2.4. Virulence of C. rodentium to Mice Is Attenuated by ompA Gene Deletion

To examine T3SS-associated virulence of the *ompA* mutant, *C. rodentium* was used for an alternative intestinal pathogen. EHEC is a human-specific pathogen that does not cause typical diarrhea symptoms in mice while *C. rodentium* is a mouse pathogen that produces a subset of orthologous T3SS proteins, including EspA and EspB, but it does not have genes that encode Shiga toxins [26]. Thus, *C. rodentium* is often used to evaluate T3SS-associated virulence in mice. The *ompA* deletion mutant was derived from the *C. rodentium* DBS100 strain (Figure 7A) The DBS100 strain is highly virulent to C3H/HeJ mice. All mice infected with this strain died within 10 days. However, the mice infected with the *ompA* mutant survived significantly longer, with all infected mice dying within 12 days post-infection (Figure 7B). Moreover, diarrhea symptoms in mice infected with the *ompA* mutant developed more slowly than in those infected with the parent strain (Figure 7C). These results indicate that bacteria require OmpA to develop optimal virulence in mice.

We suspected that the *ompA* mutant is more susceptible to acid than the parent strain, and its defective virulence is due to its low ability to survive the acidic conditions of the gastric transit after oral infection. However, after growing both the parent strain and *ompA* mutant in an acidified medium (pH 3.5), we found no significant difference in their survival rates (survival rates: 25.0 ± 0.8% and 26.5 ± 0.7% for the DBS100 parent strain and the *ompA* mutant of DBS100, respectively; 34.9 ± 1.7% and 31.5 ± 2.6% for the EHEC parent strain and the *ompA* mutant of EHEC, respectively) (Figure 8). Therefore, the defective virulence of the *ompA* mutant is not due to a higher susceptibility to acid.

We also suspected that the *ompA* mutant is more susceptible to bile salts than the parent strain, and its ability to survive in the intestinal tract is low, which may explain the attenuated virulence of the *ompA* mutant. To test this hypothesis, we compared the susceptibilities of the *ompA* mutant and the parent strain to bile salts. Compared to the parent strains, the *ompA* mutants of *C. rodentium* DBS100 and EHEC exhibited lower MICs in response to sodium cholate and sodium deoxycholate (Table 2).

## 3. Discussion

The diverse functions of OmpA include its role as a gating channel and a receptor for bacteriophages, and its roles in maintaining bacterial surface integrity and in biofilm formation [27,28,29,30]. It is also involved in bacterial pathogenesis. In *E. coli*, OmpA contributes to bacterial pathogenesis through its involvement in the adherence of EHEC to the intestinal cells, the colony formation of UPEC in the urinary tract of mice, the invasion of meningitis strains into brain microvascular cells, and the impairing of immune cells [12,13,14,15]. Our results showed that compared to the parent strain, an *ompA* deletion mutant secreted less type III secretory proteins (EspA and EspB) and formed A/E lesions and lysed RBCs with lower degrees, suggesting that OmpA sustains the T3SS activity in EHEC (Figure 2, Figure 3, Figure 5 and Figure 6). We also showed that deletion of *ompA* in *C. rodentium* prolonged the survival period of mice by retarding the development of diarrhea symptoms (Figure 7). This supports the results of in vitro experiments using EHEC. *C. rodentium* has been well defined as an alternative pathogen to evaluate virulence associated with the T3SS activity in the intestine of mice because *C. rodentium* produces a set of T3SS protein orthologs that share high sequence similarity with that in EHEC, while it does not produce Shiga toxins, and T3SS-deficient strains did not exhibit any virulence phenotypes in mice [25,26] Therefore, we believe that OmpA contributes to T3SS-associated virulence in EHEC. The *ompA* mutant was more susceptible to bile salts than the parent strain (Table 2). Deletion of *ompA* results in decreased adhesion of the bacteria to intestinal epithelial cells [14]. During the process of infection, EHEC initially adheres to intestinal epithelial cells, then it produces a subset of T3SS proteins (i.e., effector proteins), which it then injects into host cells via transport machinery [3,4]. Therefore, bacterial adhesion to host cells is a critical step that enables EHEC to induce virulence associated with the T3SS. In addition to reduced T3SS activity, increased susceptibility to bile acids and decreased adhesion to intestinal epithelial cells may also explain the attenuated virulence of the *ompA* mutant in mice.

OmpA, OmpW, and OmpX have common eight-stranded β-barrel structures although they do not share sequence homology [11]. However, in contrast to *ompA* deletion, deletion of *ompW* and *ompX* did not decrease EspB levels (Figure 2). *P. aeruginosa* and some Vibrio species produce high levels of OmpW orthologs. In *P. aeruginosa*, a mutant lacking OmpW is less cytotoxic to human bronchial epithelial cells, while a similar *V. cholerae* mutant is unable to colonize in mice [19,20]. Meanwhile, the OmpW protein is considered a minor outer membrane protein in *E. coli* because its production level is relatively low compared to other major outer membrane proteins, such as OmpA [31]. This may explain why the *ompW* deletion did not affect EspB secretion. Similarly to OmpA, OmpX is highly expressed in *E. coli*, and both proteins share highly conserved transmembrane domain structures [32]. However, unlike OmpA, OmpX lacks a periplasmic domain, which may help determine the roles of these proteins play in pathogenicity. The distinctive roles of OmpA and OmpX in pathogenesis have been observed in UPEC. On the one hand, deletion of *ompA* impairs colony formation in the urinary tract of mice, although the *ompA* mutant still retains the ability to aggregate within bladder epithelial cells [15,33]. On the other hand, deletion of *ompX* results in defective bacterial aggregation with reduced colonization in the urinary tract of mice [17]. 

OmpA is known to be important for outer membrane stability [29,34]. In the protein complex comprising the T3SS, EscC is required to form the “outer ring” embedded in the outer membrane, and it participates in forming the needle-shaped transport protein complex [35,36]. Deleting the *ompA* gene may impair the precise localization of the “outer ring” proteins and stability of the transport protein complex, which, in turn, negatively impacts protein secretion activity. We suggest that OmpA is required for the activity of the T3SS, and for optimal virulence in EHEC. Overall, our results provide additional insight into the role of OmpA in bacterial pathogenesis.

## 4. Materials and Methods

### 4.1. Bacterial Strains, Host Cells, and Culture Conditions

The bacterial strains and plasmids used in this study are listed in Table 3. Unless otherwise indicated, bacteria were grown in Luria–Bertani (LB) medium, and cell growth was monitored by measuring optical density at 600 nm (OD_600_). The following antibiotics were added to the growth media for marker selection and plasmid maintenance: 45 μg/mL chloramphenicol and 50 μg/mL kanamycin. HeLa cells were cultured in Dulbecco’s Modified Eagle Medium (DMEM) containing 10% HyClone Fetal Clone III serum (HyClone Laboratories, Inc., Logan, UT, USA).

### 4.2. Cloning and Mutant Construction

In-frame gene deletions were produced by sequence overlap extension PCR using a strategy described previously [37], and the primer pairs, delta1/delta2 and delta3/delta4 primers, are shown in Table 4. The upstream flanking DNA comprised 450 bp and the first three amino acid codons for *ompA* and *ompW* and the first four amino acid codons for *ompX*. The downstream flanking DNA comprised the last two amino acid codons for *ompA* and *ompX*, the last three amino acid codons for *ompW*, the stop codon, and 450 bp of DNA. This deletion construct was ligated into the temperature-sensitive plasmid pKO3 [37], and the resulting plasmids were introduced into EHEC and *C. rodentium* DBS100 strains. We selected sucrose-resistant/chloramphenicol-sensitive colonies at 30 °C. We constructed an OmpA expression plasmid, pTrc99KompA, by amplifying the *ompA* gene using primer pairs (listed in Table 4), then ligating the PCR product into the pTrc99K vector [38]. All constructs were confirmed by DNA sequencing.

### 4.3. Western Blotting

EHEC strains were cultured to the early stationary phase in DMEM at 37 °C. Extracellular proteins were precipitated from culture supernatants using 10% trichloroacetic acid (TCA) and then dissolved in Laemmli sample buffer (Bio-Rad Laboratories, Hercules, CA, USA). Bovine serum albumin (BSA) was used as a loading control and was added to the extracellular protein samples prior to the precipitation with TCA. Intracellular proteins were extracted by resuspending the bacterial cells in 50 mM phosphate buffer containing 8 M urea and then lysing the cells by sonication. Intracellular proteins (12 μg) and extracellular proteins were separated on a 12.5% acrylamide Tris-glycine sodium dodecyl sulfate-polyacrylamide gel electrophoresis (SDS-PAGE) gel, respectively. The gel was electroblotted onto a polyvinylidene fluoride membrane (Bio-Rad Laboratories, Hercules, CA, USA). EspA and EspB were detected with EspA and EspB antisera, respectively, which were in turn detected with anti-rabbit horseradish peroxidase-conjugated immunoglobulin G secondary antibody (Sigma-Aldrich Co. LLC., St. Louis, MO, USA) and a SuperSignal West Pico Kit (Thermo Fisher Scientific, Waltham, MA, USA) [25,39]. Protein bands were analyzed on an LAS-4000 Luminescent Image Analyzer equipped with the Image Quant LAS 4000 software (GE Healthcare Japan, Tokyo, Japan). 

### 4.4. Shiga Toxin Assay

The levels of Shiga toxins (Stx1 and Stx2) were measured using latex agglutination reagents (Denka Seiken Co. Ltd., Tokyo, Japan). EHEC strains were grown at 37 °C with shaking in Muller–Hinton medium until they reached early stationary phase. These culture supernatants were serially diluted in 96-well round bottomed plates containing phosphate-buffered saline (PBS). The latex suspension sensitized with the Stx1 or Stx2 antibody was then added. The plates were incubated for 14 h at 4 °C, and then the titers were determined by the reciprocal of the last dilution before agglutinations were observed.

### 4.5. RNA Extraction and Quantitative Real-Time PCR

We grew bacteria to the late-logarithmic growth phase (OD_600_ ~ 0.5) and then carried out total RNA extraction and real-time PCR as previously described [25]. The constitutively expressed *rrsA* and *rpoD* genes were used as internal controls. The primers used for real-time PCR are listed in Table 5.

### 4.6. A/E Lesion Formation

EHEC strains were statically grown in DMEM overnight at 37 °C under an atmosphere of 5% CO_2_, and inoculated into HeLa cells with an MOI of 200 bacterial cells per host cell, and incubated for 3 h at 37 °C under an atmosphere of 5% CO_2_. The medium was replaced by a fresh one, then the culture was continued for another 2 h. Actin and bacteria/HeLa nuclei were stained with Rhodamine phalloidin (Life technologies, Carlsbad, CA, USA) and Hoechst33342 (Dojindo Laboratories, Tokyo, Japan), respectively. Fluorescent images were acquired in the DAPI and rhodamine phalloidin laser units on an Olympus FV 1200 IX81 microscope (Olympus Corporation, Tokyo, Japan).

### 4.7. Hemolysis Assay

T3SS-dependent hemolytic activity was assessed as described previously [40]. EHEC strains were statically grown in DMEM overnight at 37 °C under an atmosphere of 5% CO_2_. In brief, human red blood cells (RBCs) and bacteria were resuspended to 3 × 10^9^ cells/mL and 7 × 10^9^ cells/mL with DMEM, respectively. Fifty microliters of RBC suspension and 50 μL of bacterial suspension were added to a flat bottom 96-well tissue culture plate. An RBC suspension without bacteria was used as a negative control, and RBCs lysed with distilled water were used as a positive control. After centrifugation to tightly attach bacteria to the RBCs, the plate was incubated at 37 °C under an atmosphere of 5% CO_2_. At the indicated time, 100 μL of DMEM was added to each well. After centrifugation, 100 μL of supernatant was carefully collected onto an ELISA plate. The OD_450_ was measured. Percent hemolysis was calculated as follows: {(sample) − (negative control)}/{(positive control) − (negative control)} × 100.

### 4.8. C. rodentium Infections in Mice

*C. rodentium* DBS100 or its *ompA* mutant was aerobically grown overnight in LB medium at 37 °C. The bacterial cells were harvested and resuspended in fresh LB medium at a concentration of 1 × 10^9^ CFU/mL, and 200 μL of the bacterial suspension (2 × 10^8^ CFU) was orally administrated to four-week-old female C3H/HeJ mice (CLEA Japan, Tokyo, Japan) (*N* = 5 mice for each group). In the control group, 3 mice were inoculated with 200 μL of bacteria-free LB broth. The mice were monitored daily for 14 days to determine survival rates and to obtain stool scores. Briefly, the daily stool samples were frozen at −20 °C until the end of the experiments. These were then thawed, and stool scores were determined using scoring criteria derived from a previous study [41] (0, normal; 1, loose stool; 2, shapeless loose stool; 3, diarrhea with brown color; 4, watery diarrhea with almost white or no color). All stool samples were compared to those of the uninfected controls.

### 4.9. Bile Salts Susceptibility Assays

The susceptibility of bacteria to bile salts was tested by determining the minimum inhibitory concentrations (MICs) of sodium deoxycholate and sodium cholate as previously described [25].

### 4.10. Acid Survival Assays

Bacteria were grown to the late logarithmic phase in LB medium at 37 °C. The culture was diluted into acidified (pH 3.5, adjusted with HCl) or regular LB medium (pH 7.2) and incubated for 1 h. The percentage survival was calculated as the number of CFUs generated after incubation in the acidified LB medium relative to the number of CFUs generated after incubation in the regular LB medium.

### 4.11. Statistical Analysis

We used the Gehan–Breslow–Wilcoxon tests for mouse survival experiments and the unpaired *t*-tests for hemolysis assays, mouse stool scores, and acid challenge experiments, and then determined *p*-values using GraphPad Prism version 6.00.

### 4.12. Ethics Information

All animal studies were approved by the Committee of Experimental Animal Research of Gunma University (approval number: 19-094). All experiments were conducted in accordance with the guidelines and regulations.

## 5. Conclusions

This study showed that the β-barrel outer membrane protein OmpA sustains the activity of the T3SS, and is required for optimal virulence in EHEC.

## Figures and Tables

**Figure 1 pathogens-10-01496-f001:**
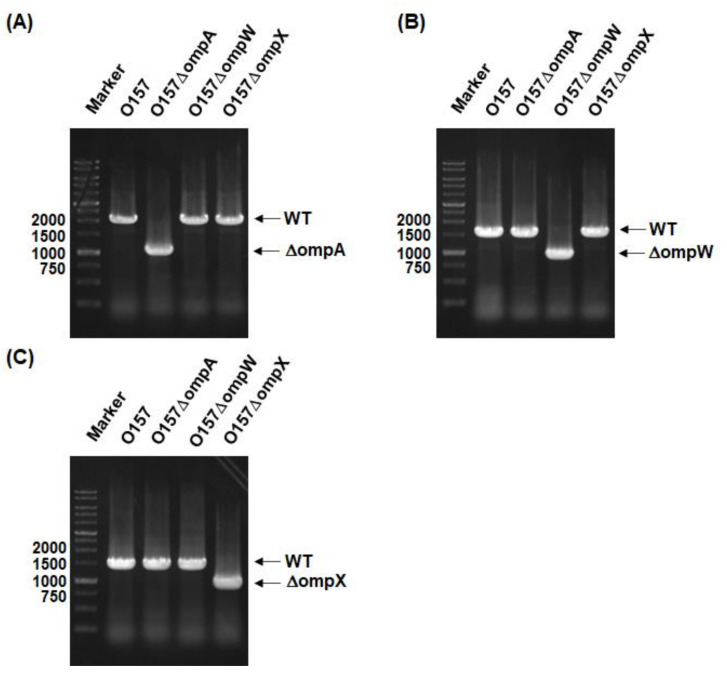
Confirmation of *ompA*, *ompW*, and *ompX* gene deletion constructs. Gene deletions of (**A**) *ompA*, (**B**) *ompW*, and (**C**) *ompX* from the O157 parent strain were confirmed by PCR assays using each delta1/delta4 primer pair shown in the Materials and Methods section. Locations of molecular markers (in base pairs) are shown on the left. The predicted sizes of amplified DNA fragments from the (**A**) *ompA*, (**B**) *ompW*, and (**C**) *ompX* intact strains are approximately1950, 1550, and 1430 base pairs, respectively, while those from deletion mutants are approximately 900 base pairs.

**Figure 2 pathogens-10-01496-f002:**
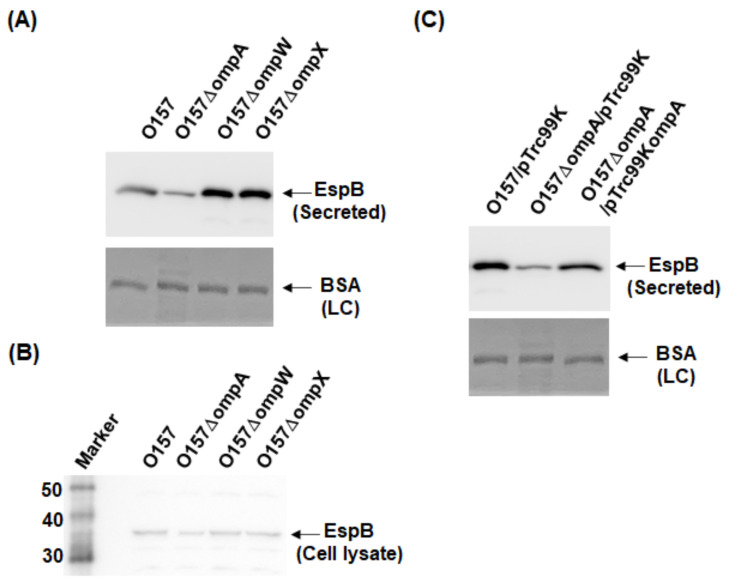
Extracellular (secreted) and intracellular (cell lysate) EspB levels in the O157 parent strain and its derivatives. (**A**,**B**) The parent strain and the *ompA*, *ompW*, and *ompX* mutants were grown, then we determined extracellular (**A**) and intracellular (**B**) EspB levels. (**C**) We cultured the parent strain and *ompA* mutant carrying pTrc99K or pTrc99KompA in the presence of 0.01 mM IPTG. EspB was visualized by western blotting with EspB antiserum. For loading control (LC), BSA was visualized by CBB staining.

**Figure 3 pathogens-10-01496-f003:**
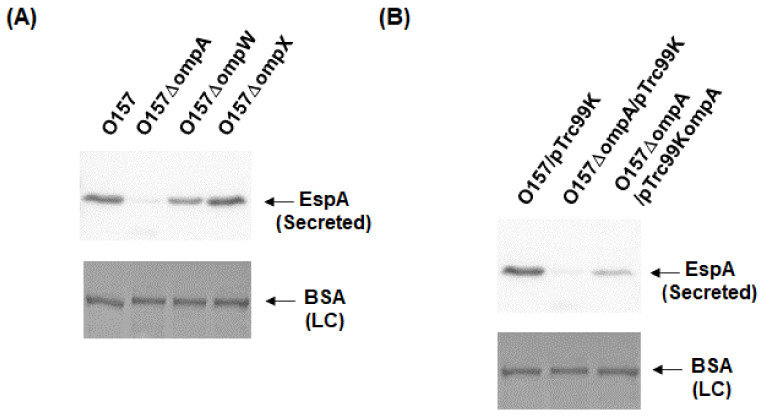
Extracellular (secreted) EspA levels in the O157 parent strain and the *ompA* mutant. (**A**) The parent strain and the *ompA* mutant were grown, then we determined extracellular EspA levels. (**B**) We cultured the parent strain and *ompA* mutant carrying pTrc99K or pTrc99KompA in the presence of 0.01 mM IPTG. EspA was visualized by western blotting with EspA antiserum. For loading control (LC), BSA was visualized by CBB staining.

**Figure 4 pathogens-10-01496-f004:**
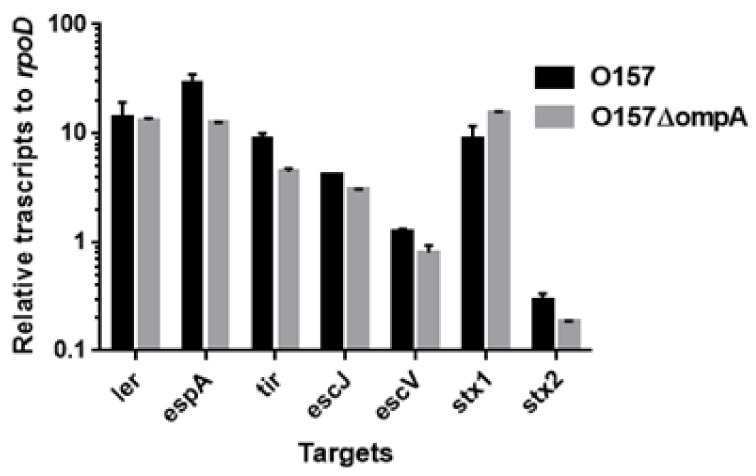
Transcript levels of LEE and *stx* genes in the O157 parent strain and the *ompA* mutant. Transcript levels were represented as relative values to that of *rpoD* (housekeeping gene). Data plotted are the means of two biological replicates, error bars indicate the ranges.

**Figure 5 pathogens-10-01496-f005:**
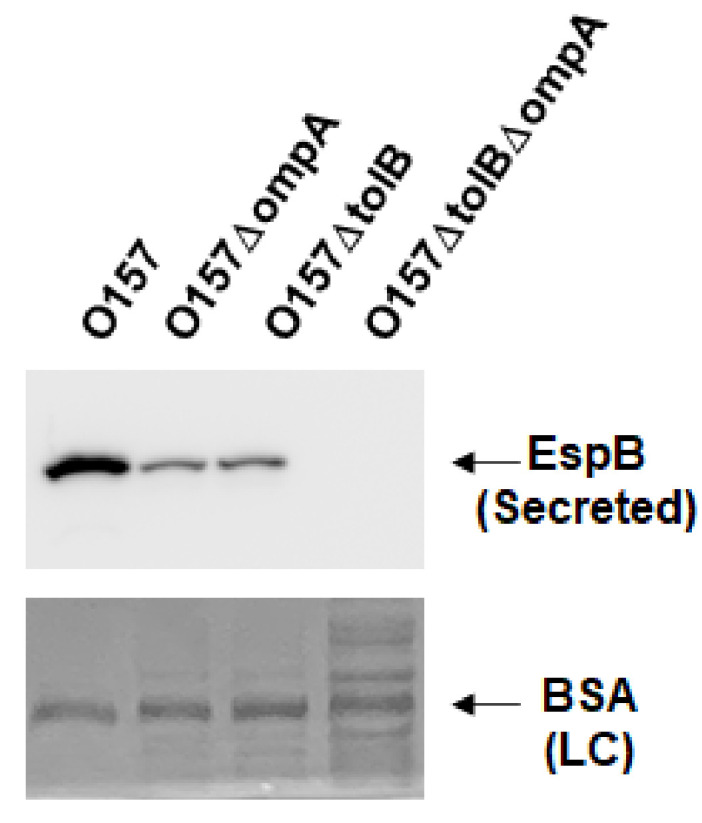
Extracellular (secreted) EspB levels in the O157 parent strain and the *ompA* and *tolB* mutants. The parent strain and the *ompA* and *tolB* mutants were grown, then we determined extracellular EspB levels. EspB was visualized by western blotting with EspB antiserum. For loading control (LC), BSA was visualized by CBB staining.

**Figure 6 pathogens-10-01496-f006:**
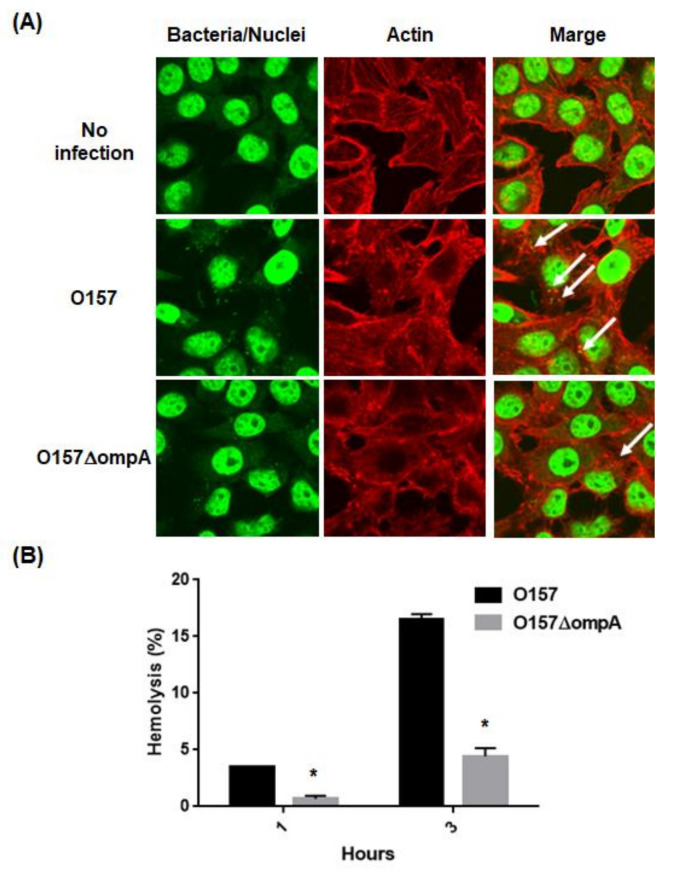
T3SS biological activities in the parent strain and *ompA* mutant. (**A**) A/E lesion formation in HeLa cells infected with the parent strain and *ompA* mutant. Bacteria and nuclei of HeLa cells stained with Hoechst33342 and actins strained with rhodamine-phalloidin were imaged, respectively, as green and red colors in the microscopy images using 100× objective. A/E lesions were indicated by white arrows as regions where actin was accumulated with bacteria. (**B**) T3SS-dependent RBCs hemolysis. Hemolysis (%) was represented as percent values of lysed RBCs (see Section 4). Data plotted are the means; error bars indicate the standard deviations. Asterisks denote significance for values of percent hemolysis for *ompA* mutant relative to those for the parent strain (*p* < 0.05).

**Figure 7 pathogens-10-01496-f007:**
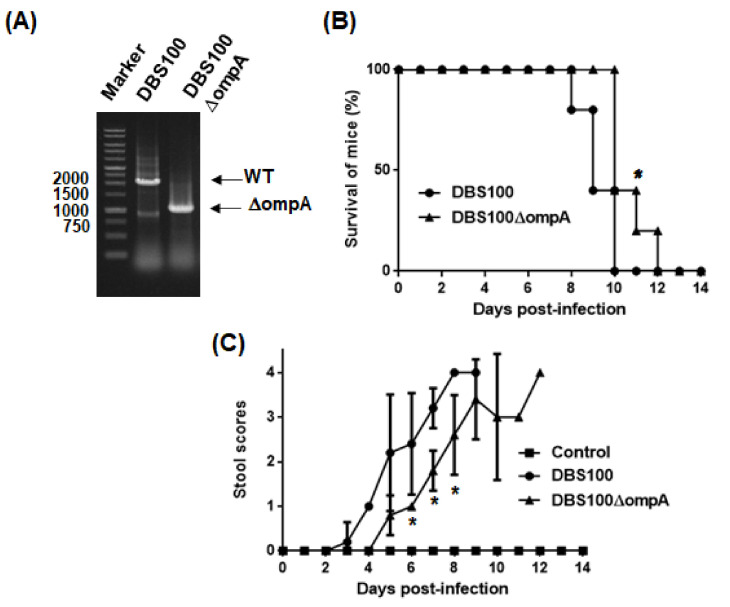
Virulence of the *C. rodentium* parent strain and the *ompA* mutant to C3H/HeJ mice. (**A**) Deletion of *ompA* from the DBS100 strain was confirmed by a PCR assay. Locations of molecular markers (in base pairs) are shown on the left. Sizes of amplified DNA fragments from the DBS100 strain and its *ompA* mutant are approximately 1950 and 900 base pairs, respectively. Mice were orally injected with the *C. rodentium* DBS100 strain or the *ompA* mutant. The mice (*N* = 5 mice per strain for parent and the *ompA* mutant, and *N* = 3 mice for non-infection control) were monitored daily. (**B**) Survival rates and (**C**) stool consistency of mice received with the parent strain or the *ompA* mutant. Stool scores were determined as the scoring criteria (0, normal; 1, loose stool; 2, shapeless loose stool; 3, diarrhea with brown color; 4, watery diarrhea with almost white or no color) by means of comparison with stools of uninfected control C3H/HeJ mice. The connecting lines denote the mean and error bars denote the standard deviation for the data. Asterisks denote significance for values of survival rate and stool scores of mice infected with *ompA* mutant relative to those infected with the parent strain (*p* < 0.05).

**Figure 8 pathogens-10-01496-f008:**
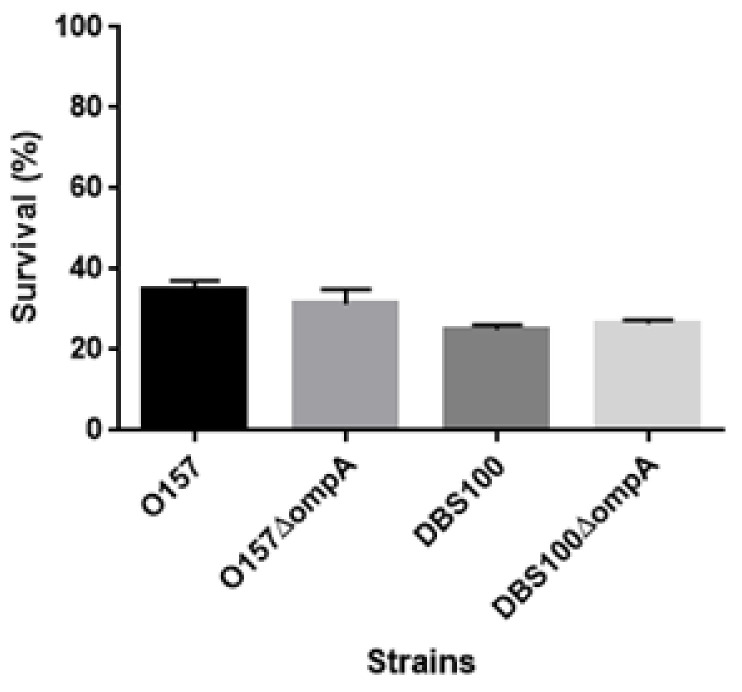
Susceptibility of O157 EHEC, DBS100 *C. rodentium* strains and their *ompA* mutants to acid. Bacterial survival was represented by percent (%) of CFU values of cells after incubation in acidified LB medium (pH3.5) relative to those of cells after incubation in the regular LB medium. Data plotted are the means; error bars indicate the standard deviations.

**Table 1 pathogens-10-01496-t001:** Shiga toxin titers of EHEC and its derivatives.

Strains	Shiga Toxin Titers	
	Stx1	Stx2
O157	128	256
O157ΔompA	128	256
O157ΔompW	128	256
O157ΔompX	128	256

**Table 2 pathogens-10-01496-t002:** Bile salt MICs for *C. rodentium* and EHEC, and its *ompA* mutants.

Strain	MICs (mg/L) of	
	Sodium Deoxycholate	Sodium Cholate
O157	>51,200	>51,200
O157ΔompA	51,200	>51,200
DBS100	25,600	51,200
DBS100ΔompA	1600	6400

**Table 3 pathogens-10-01496-t003:** Strains and plasmids used in this study.

Strains or Plasmids	Relevant Genotype/Phenotype	Reference
Strains		
O157	EHEC O157:H7 [RIMD0509952]	RIMD 0509952
O157ΔompA	*ompA* mutant	this work
O157ΔompW	*ompW* mutant	this work
O157ΔompX	*ompX* mutant	this work
O157ΔtolB	*tolB* mutant	[25]
O157ΔtolBΔompA	*tolB* and *ompA* double mutant	this work
DBS100	*Citrobacter rodentium* (ATCC 51459)	ATCC51459
DBS100ΔompA	*ompA* mutant	this work
Plasmids		
pKO3	Temperature-sensitive vector for gene targeting, *sacB*, Cm^R^	[37]
pTrc99K	Vector for IPTG-inducible expression; Km^R^	[38]
pTrc99KompA	*ompA* expression plasmid; Km^R^	this work

Cm^R^: chloramphenicol resistance, Km^R^: kanamycin resistance.

**Table 4 pathogens-10-01496-t004:** Primers used for plasmid construction.

Primers	DNA Sequence (5′—3′)	Use
ompA-delta1	GCGGGATCCTTTGACTGCAGAAGAGCATGC	O157ΔompA construction
ompA-delta2	CCAGACGAGAACTTAAGCCTGCTTTTTCATTTTTTGCGCCTCG	O157ΔompA construction
ompA-delta3	CGAGGCGCAAAAAATGAAAAAGCAGGCTTAAGTTCTCGTCTGG	O157ΔompA construction
ompA-delta4	GCGGTCGACAGCGGTTGGAAATGGAAGTATC	O157ΔompA construction
ompW-delta1	GCGGGATCCACGCACATAGCAACGATACC	O157ΔompW construction
ompW-delta2	TGCAGAGAAAATTAAAAACGATACTTTTTCATATCCGCTCCGTC	O157ΔompW construction
ompW-delta3	CGACGGAGCGGATATGAAAAAGTATCGTTTTTAATTTTCTCTGC	O157ΔompW construction
ompW-delta4	GCGGTCGACGTTATCTTCATGTCGAACAGC	O157ΔompW construction
ompX-delta1	GCGGGATCCGACTTAGCTAACGAGGCTCC	O157ΔompX construction
ompX-delta2	ATATCACCGAAGTGATTAGAAGCGAATTTTTTTCATAACCACCTC	O157ΔompX construction
ompX-delta3	TTTGAGGTGGTTATGAAAAAAATTCGCTTCTAATCACTTCGGTG	O157ΔompX construction
ompX-delta4	GCGGTCGACAAACAGACGATTTACTGCGC	O157ΔompX construction
ompA-CR-delta1	GCGGGATCCGTTAACGGAAGAAGAACACGC	DBS100ΔompA construction
ompA-CR-delta2	GACGGAAACTTAAGCCTGCGGCTTTTTCATTTTTTGCGCCTCG	DBS100ΔompA construction
ompA-CR-delta3	CGAGGCGCAAAAAATGAAAAAGCCGCAGGCTTAAGTTTCCGTC	DBS100ΔompA construction
ompA-CR-delta4	GCGGTCGACAGCGGTTGGAAATGGAAGTATC	DBS100ΔompA construction
pTrcompA-F	GCGCCATGGAAAAGACAGCTATCGCG	pTrc99KompA construction
pTrcompA-R	GCGGTCGACTTAAGCTTGCGGCTGAGTTAC	pTrc99KompA construction

**Table 5 pathogens-10-01496-t005:** Primers used for real-time PCR analyses.

Primer	DNA Sequence (5′—3′)	Use
rrsA-qPCR-F	CGGTGGAGCATGTGGTTTAA	Quantitative real-time PCR
rrsA-qPCR-R	GAAAACTTCCGTGGATGTCAAGA	Quantitative real-time PCR
rpoD-qPCR-F	CAAGCCGTGGTCGGAAAA	Quantitative real-time PCR
rpoD-qPCR-R	GGGCGCGATGCACTTCT	Quantitative real-time PCR
ler-qPCR-F	CGACCAGGTCTGCCCTTCT	Quantitative real-time PCR
ler-qPCR-R	TCGCTCGCCGGAACTC	Quantitative real-time PCR
espA-qPCR-F	CCGTTGTCAGGTTATTCGCTTT	Quantitative real-time PCR
espA-qPCR-R	TGATTTAAGCGCTGGTGATCTG	Quantitative real-time PCR
tir-qPCR-F	TTTTTGCGCCTGAGCATTATT	Quantitative real-time PCR
tir-qPCR-R	GCTAAAGCAGCAGGCGAAGA	Quantitative real-time PCR
escJ-qPCR-F	AAAGAAGCTAATCAGATGCAAGCA	Quantitative real-time PCR
escJ-qPCR-R	TCCACTTTTGTCCATTTCTTTGG	Quantitative real-time PCR
escV-qPCR-F	CGCCTGCGGTGACAGAA	Quantitative real-time PCR
escV-qPCR-R	TTGTCTGTCGGTGATGCTTTG	Quantitative real-time PCR
stx1-qPCR-F	TCGCGAGTTGCCAGAATG	Quantitative real-time PCR
stx1-qPCR-R	TTCCATCTGCCGGACACAT	Quantitative real-time PCR
stx2-qPCR-F	TTCGCGCCGTGAATGAA	Quantitative real-time PCR
stx2-qPCR-R	CAGGCCTGTCGCCAGTTATC	Quantitative real-time PCR

## Data Availability

The data presented in this study are available on request from the corresponding author.

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
