# Peer review of "Roles of OmpA in Type III Secretion System-Mediated Virulence of Enterohemorrhagic Escherichia coli"

_pathogens, 2021, doi:10.3390/pathogens10111496_

Round 1

Reviewer 1 Report

This study investigated the influence of ompA deletion on the secretion of T3SS translocators and the pathogenicity of EHEC. A key finding of this study is that the authors show ompA deletion affects the secretion but not the expression of EspA and EspB. However, the data presented in this study are not enough to give a conclusion that OmpA has a novel role in the pathogenesis of EHEC.

Major concerns:

  1. The authors showed that the deletion of ompA affects the secretion of EspA and EspB. However, they did not give further evidence to show whether the function of the T3SS was also disrupted.
  2. As OmpA has been previously shown to be critical for surface integrity, serum resistance, adhesion, invasion, etc., why the authors believe that the attenuated virulence of the ompA mutant observed in this study was attributed to the influence on T3SS? More and direct evidence is needed to draw this conclusion.
  3. How does OmpA deletion affect T3SS?
  4. Line 146-154, the author tested the influence of ompA deletion on susceptibility to bile salts. Why this part is included here? What’s the relationship between bile salts susceptibility and T3SS?

Minor:

  1. In Fig. 1 and Fig. 2, there should be a control protein for the secreted samples as well as the whole-cell samples to show similar amounts of protein were loaded.
  2. line 174-176, this conclusion is not appropriate.

Author Response

This study investigated the influence of ompA deletion on the secretion of T3SS translocators and the pathogenicity of EHEC. A key finding of this study is that the authors show ompA deletion affects the secretion but not the expression of EspA and EspB. However, the data presented in this study are not enough to give a conclusion that OmpA has a novel role in the pathogenesis of EHEC.

Response: Thank you for your comments and suggestions. According to your suggestions, we revised our manuscript. In order to provide additional evidence of OmpA involvement in the T3SS-associated EHEC pathogenesis, we performed A/E lesion assay and hemolysis assay. A/E lesion assay is well recognized as a method to evaluate the T3SS activity. We found that the ability of ompA mutant to form A/E lesions is low (Fig. 6A). Hemolysis assay using RBCs is also often used to evaluate the T3SS activity. We found that hemolytic activity of the ompA mutant is lower than that of the parent strain (Fig. 6B). From these results, we believe that our conclusion can be justified.

Major concerns:

1. The authors showed that the deletion of ompA affects the secretion of EspA and EspB. However, they did not give further evidence to show whether the function of the T3SS was also disrupted.

Response: As mentioned above, the ompA mutant exhibited less A/E lesion formation and lower hemolytic activity than the parent strain (Fig 6). From these results, we believe that the T3SS activity is impaired by the deletion of ompA.

2. As OmpA has been previously shown to be critical for surface integrity, serum resistance, adhesion, invasion, etc., why the authors believe that the attenuated virulence of the ompA mutant observed in this study was attributed to the influence on T3SS? More and direct evidence is needed to draw this conclusion.

Response: We understand that disturbance of surface integrity and reduced serum resistance, adhesion and invasion are important for bacterial pathogenesis. Previous EHEC works showed that ompA mutant exhibits a lower adhesion to the epithelial cells (We cited it as the reference in the text). Other studies on non-EHEC strains showed that ompA mutant exhibits an impaired surface integrity and reduced serum resistance and invasion. Main finding in our study is that OmpA also contributes to the T3SS activity, then deletion of ompA attenuates EHEC virulence, which is associated with reduction of the T3SS activity. We provided additional evidence that the T3SS activity is reduced by deletion of ompA (Fig. 6). Intestinal infection with C. rodentium to mice is well recognized as a model to evaluate the T3SS-dependent pathogenesis because T3SS deficient strains do not cause diarrhea in mice. It has been shown by many scientists (We also previously showed that the T3SS-deficient strain do not cause diarrhea [Ref25]). Therefore, we believe that our conclusion is scientifically fair. However, we added some discussion to describe our finding and suggestions, more clearly.

3. How does OmpA deletion affect T3SS?

Response: We have no exact answer. In addition to the data of western-blotting, we provided the data of A/E lesion and hemolysis assay (Fig. 6). From these results, we believe that ompA deletion reduces the T3SS activity. However, we do not know the mechanism. As mentioned in Discussion, ompA deletion impairs the outer ring proteins in the transport protein complex, which may reduce the T3SS activity. We think that deep structural analyses of the T3SS in the ompA mutant may be necessary. We would like to study it in our future subject.

4. Line 146-154, the author tested the influence of ompA deletion on susceptibility to bile salts. Why this part is included here? What’s the relationship between bile salts susceptibility and T3SS?

Response: We think that there is no relationship between bile salts susceptibility and T3SS. However, we think that bile salts susceptibility is involved in intestinal pathogenesis because bacteria must be efficiently survived to establish the infection within intestinal sites where high concentration of bile salts is present.

From this reason, we would like to test bile acids susceptibility in addition to the T3SS activity. Now, we realize that it should be discussed in the mice experiment. Therefore, the description of bile salts susceptibility was included into mice experiment.

Minor:

1. In Fig. 1 and Fig. 2, there should be a control protein for the secreted samples as well as the whole-cell samples to show similar amounts of protein were loaded.

Response: As suggested, we included the BSA protein as a loading control (Fig. 2, 3 and 5). The BSA protein was added into supernatant samples prior to TCA addition. The BSA protein was precipitated with extracellular proteins including EspA and EspB.

2. line 174-176, this conclusion is not appropriate.

Response: As mentioned above, we added the data of A/E lesion and hemolysis assay (Fig. 6). We showed that the abilities of A/E lesion formation and RBC hemolysis associated with the T3SS activity are reduced by ompA deletion. We believe that this conclusion is now appropriate.

Reviewer 2 Report

In this manuscript entitled “Roles of the b-barrel, small-sized outer membrane protein, OmpA, in Type III secretion system-mediated virulence of enterohemorrhagic Escherichia coli”, the authors suggest that OmpA is required for the activity of the Type III secretion system and for optimal virulence in enterohemorrhagic Escherichia coli and Citrobacter rodentium.

-In general, the manuscript needs a revision of the English and the narrative.

-The title should be simplified and commas should be reduced.

-The abstract has excessive use of parentheses. Rewrite lines 14-20, they are repetitive.

-Note that the conversion of the file to .pdf format changes the symbology (beta and micro).

-Type three secretion system should be named only once in full and then replaced in the text by "T3SS".

-References should be in square brackets and in black color. When there is more than one reference they should be grouped together.

-Subtitles should be numbered.

-line 43: incorporate "translocated intimin receptor (Tir) is the first T3SS effector delivered into the host cell".

-line 45-50: include EspD as part of the T3SS translocon.

-line 57-59: missing reference

-Incorporate results showing that ompA, ompX and ompW mutants do not express these proteins. At least PCR assays showing deletion of the genes.

-Line 117: Separate EspA-B and Stx assays with a subheading.

-It would be interesting to include an in vitro T3SS activity assay (red blood cell hemolysis assay, effector translocation assays to epithelial cells, pedestal formation, etc). In this way it could be demonstrated that the absence of OmpA affects T3SS formation and activity.

-Incorporate graphs showing that Citrobacter rodentium delta ompA strain shows reduced secretion of EspA and EspB in vitro (western blot assays).

Complete: Author Contributions

Author Response

In this manuscript entitled “Roles of the b-barrel, small-sized outer membrane protein, OmpA, in Type III secretion system-mediated virulence of enterohemorrhagic Escherichia coli”, the authors suggest that OmpA is required for the activity of the Type III secretion system and for optimal virulence in enterohemorrhagic Escherichia coli and Citrobacter rodentium.

Response: Thank you for your comments and suggestions. According to your suggestions, we revised our manuscript. As suggested by you and the other reviewer, we provided some data including PCR assays, a loading control for western-blotting, A/E lesion formation and RBC hemolysis assay.

-In general, the manuscript needs a revision of the English and the narrative.

Response: Our manuscript was checked again by a commercial English editing service (Crimson Interactive Japan Co., Ltd.).

-The title should be simplified and commas should be reduced.

Response: As suggested, we simplified the title (The changed title is “Roles of OmpA in Type III secretion system-mediated virulence of enterohemorrhagic Escherichia coli”).

-The abstract has excessive use of parentheses. Rewrite lines 14-20, they are repetitive.

Response: As suggested, we decreased use of parentheses, and line14-20 was revised as “Outer membrane proteins are commonly produced by gram-negative bacteria, and they have diverse functions. A subgroup of proteins, which includes OmpA, OmpW and OmpX, is often involved in bacterial pathogenesis.”.

-Note that the conversion of the file to .pdf format changes the symbology (beta and micro).

Response: Corrected. Thank you for catching.

-Type three secretion system should be named only once in full and then replaced in the text by "T3SS".

Response: As suggested, we revised it through the text.

-References should be in square brackets and in black color. When there is more than one reference they should be grouped together.

Response: We accordingly modified them.

-Subtitles should be numbered.

Response: We accordingly added numbers.

-line 43: incorporate "translocated intimin receptor (Tir) is the first T3SS effector delivered into the host cell".

Response: We added it (Tir is delivered into the host cells as the first effector, and re-translocated onto the host cell membrane, which enables bacteria to attach the host cells [5].).

-line 45-50: include EspD as part of the T3SS translocon.

Response: We added it.

-line 57-59: missing reference

Response: We added references.

-Incorporate results showing that ompA, ompX and ompW mutants do not express these proteins. At least PCR assays showing deletion of the genes.

Response: As described above, we added the data of PCR assays (Fig. 1 and 7A).

-Line 117: Separate EspA-B and Stx assays with a subheading.

Response: According to the suggestion, we separated them.

-It would be interesting to include an in vitro T3SS activity assay (red blood cell hemolysis assay, effector translocation assays to epithelial cells, pedestal formation, etc). In this way it could be demonstrated that the absence of OmpA affects T3SS formation and activity.

Response: We agree that these are important experiment. We performed A/E lesion (pedestal formation) assay and RBC hemolysis assay as suggested. We found that the ompA mutant has lower abilities of A/E lesion formation and RBC hemolysis than the parent strain (Fig 6).

-Incorporate graphs showing that Citrobacter rodentium delta ompA strain shows reduced secretion of EspA and EspB in vitro (western blot assays).

Response: Unfortunately, both of our EspA and EspB antisera did not react to proteins of C. rodentium because these antisera were produced from peptide antigens according to the amino acid sequence of the EHEC O157 Sakai strain, and the sequence regions including antigens in EHEC is unfortunately different from that in C. rodentium DBS100.

Complete: Author Contributions

Response: We added it.

Round 2

Reviewer 1 Report

Most of my concerns have been addressed. I have no further comments.

Reviewer 2 Report

I believe that you have made appropriate corrections.